# Learning rule influences recurrent network representations but not attractor structure in decision-making tasks

**Brandon McMahan**     **Michael Kleinman**     **Jonathan C. Kao**
Department of Electrical and Computer Engineering
University of California, Los Angeles
{bmcmahan2025, michael.kleinman}@ucla.edu     kao@seas.ucla.edu

## Abstract

Recurrent neural networks (RNNs) are popular tools for studying computational dynamics in neurobiological circuits. However, due to the dizzying array of design choices, it is unclear if computational dynamics unearthed from RNNs provide reliable neurobiological inferences. Understanding the effects of design choices on RNN computation is valuable in two ways. First, invariant properties that persist in RNNs across a wide range of design choices are more likely to be candidate neurobiological mechanisms. Second, understanding what design choices lead to similar dynamical solutions reduces the burden of imposing that all design choices be totally faithful replications of biology. We focus our investigation on how RNN learning rule and task design affect RNN computation. We trained large populations of RNNs with different, but commonly used, learning rules on decision-making tasks inspired by neuroscience literature. For relatively complex tasks, we find that attractor topology is invariant to the choice of learning rule, but representational geometry is not. For simple tasks, we find that attractor topology depends on task input noise. However, when a task becomes increasingly complex, RNN attractor topology becomes invariant to input noise. Together, our results suggest that RNN dynamics are robust across learning rules but can be sensitive to the training task design, especially for simpler tasks.

## 1 Introduction

Computational neuroscientists have increasingly used recurrent neural networks (RNNs) to model cortical computation and gain insight into how the brain performs cognitive and motor tasks. Typically, neuroscientists design RNNs to model a particular brain region during a behavioral task by generating an artificial dataset that abstracts the task. RNNs are trained to perform this task and often exhibit similar representations and low-dimensional dynamics to neural population recordings. This similarity is used to argue that the RNN is a good model of the brain circuit's computations and neural dynamics. In this way, RNNs have been successfully used to study cognitive tasks, including decision-making and working memory [1–7], as well as motor tasks [8–12]. Because RNN parameters are fully observed, trained RNNs can be analyzed to propose new hypotheses for how these computations are carried out in the brain [13].

One potential concern of this approach is that there are significant differences between RNNs and biological circuits. Further, RNNs may exhibit different representations and dynamics as a result of design choices, such as hyperparameters and architectures. Maheswaranathan, Williams, et al. [14] investigated whether there were any universal properties across recurrent architectures and activation functions. They found through a numerical study that RNNs tend to exhibit different representations across architectures (RNN, GRU, LSTM, UGRNN) and activation function (relu, tanh)

35th Conference on Neural Information Processing Systems (NeurIPS 2021).

but exhibited universality in attractor topologies. These results encouragingly suggest that, although RNN architecture and activation function may result in different representations, the dynamical mechanisms of these networks appear to be conserved irrespective of these choices.

A similarly important aspect of RNN modeling is how networks are trained, which differ in nature from biological neural circuits. Neural circuits are believed to employ Hebbian learning principles [15], and while it is possible that they may employ a backpropagation-like algorithm, typical training approaches in RNNs are not necessarily biologically plausible [16, 17]. For example, backpropagation-through-time assumes global knowledge of all the parameter weights in the network. Computational neuroscience studies utilize a diverse range of learning algorithms, from stochastic gradient descent with backpropagation-through-time [1–3, 5, 6, 9–11], Hebbian-inspired learning [18], FORCE learning [19, 4, 20], and evolutionary learning through genetic algorithms [21]. Different learning rules result in different learning trajectories through parameter space, leading to the possibility that RNN computation is learning rule dependent. If learning rule affects the learned solution, then conclusions of computational neuroscience studies may depend on the experimenter-chosen learning rule. We therefore sought to empirically study the similarities and differences in RNN representations and dynamics across learning rules.

In particular, we study the topological structure of RNN attractor (or fixed point) states and the representational geometry following training with different learning rules. We find that all learning rules resulted in similar attractor topologies across different decision-making tasks. However, while we find evidence that the attractor structure is invariant to the choice of learning rule, it is not invariant to task hyperparameters. We found that task hyperparameters for the contextual integration task from different studies resulted in RNNs with different attractor structure and dynamical mechanisms. Interestingly, we also find that as we train RNNs on progressively more complex tasks, the attractor topology remains invariant to the choice of learning rule even though representational geometries become more individualistic. The results of our study therefore provide insights into what properties of RNNs are likely conserved across design choices, which in turn supports the study of RNNs to probe dynamics in biological neural circuits.

## 2   Methods

### 2.1   Model Architecture

We trained continuous time RNNs, which when discretized via Euler's method, evolve as

$$\mathbf{x}_{t+1} = (1 - \alpha)\mathbf{x}_t + \alpha\left(\mathbf{W}_{\mathrm{rec}}f(\mathbf{x}_t) + \mathbf{W}_{\mathrm{in}}\mathbf{u}_t\right) \tag{1}$$

where $\mathbf{x}_t \in \mathbb{R}^{50}$ is the pre-activation state, $\mathbf{u}_t \in \mathbb{R}^M$ is the input to the RNN, $f(\cdot)$ is the activation function, and $\alpha = \Delta t/\tau$ is the Euler time step, $\Delta t$, divided by the network time constant, $\tau$. When $\alpha = 1$, the continuous time RNN simplifies to a vanilla RNN. Unless noted otherwise, the activation function is $f(\cdot) = 1 + \tanh(\cdot)$, and we define $\mathbf{r}_t = f(\mathbf{x}_t)$. We shifted the $\tanh(\cdot)$ by 1 so that its output is always positive, as a non-negative activation is typically used for computational neuroscience studies [2]. The RNN output $\mathbf{y}_t$ is a linear readout of the hidden states,

$$\mathbf{y}_t = \mathbf{W}_{\mathrm{out}}f(\mathbf{x}_t). \tag{2}$$

$\mathbf{W}_{\mathrm{rec}}$ was initialized with weights drawn from $\mathcal{N}(0, \frac{g^2}{N})$ where $g = 1$ is the network gain and $N = 50$ is the number of artificial neurons in the RNN. $\mathbf{W}_{\mathrm{in}}$ weights were initialized from $\mathcal{N}(0, 0.5)$. $\mathbf{W}_{\mathrm{out}}$ were initialized from $\mathcal{N}(0, 0.1)$. At the beginning of all trials, the initial hidden state $\mathbf{x}_0 \in \mathbb{R}^{50}$ was drawn from $\mathcal{N}(0, 0.3)$. All networks were trained using the mean-squared error (MSE) loss. The tasks and learning rules are described in the next sections. In total, we trained and analyzed 1,920 RNNs in this numerical study[1].

### 2.2   Tasks

We used three tasks based on commonly studied perceptual decision making tasks from the neuroscience literature [22, 1, 14, 18].

---

[1]Training was carried out in approximately 7 days on an RTX 2080. H and FF RNNs were trained using CPUs both locally and on AWS

**Random dots motion** (RDM): RNNs received a 1-dimensional Gaussian noise input [22]. For each trial, we drew a random number $\mu \sim U[-0.1857, +0.1857]$, which modeled the "coherence" of a traditional RDM stimulus [23, 1]. At each timestep, $t = 0, 1, \ldots, 750$ ms, the input to the network was drawn from $\mathbf{u}_t \sim \mathcal{N}(\mu, 1)$. The RNN was constrained to output $\mathbf{y}_t = 0$ at $t = 0$ and $\mathbf{y}_t = 1$ (or $\mathbf{y}_t = -1$) at $t = 750$ ms if $\mu$ was positive (negative). These input and output values were chosen to be consistent with a previously implemented context-dependent integration task, described next [1].

**Context-dependent integration** (CDI): We implemented the CDI task in the same way as [1]. RNNs received two static binary context inputs and two time-varying Gaussian noise inputs. For each trial, one context input was 0 and the other was 1, providing a one-hot encoding of which Gaussian input stream should be integrated to compute the output. For each trial, two Gaussian noise inputs where both independently sampled from $\mathcal{N}(\mu_1, 1)$ and $\mathcal{N}(\mu_2, 1)$ where $\mu_1$ and $\mu_2$ where both independently sampled from $U[-0.1857, 0.1857]$. Samples were drawn at $t = 0, 1, \ldots, 750$ ms. RNNs were trained to output zero at $t = 0$ and $\pm 1$ at $t = 750$ ms depending on the sign of $\mu$ for the relevant input.

**N-input context-dependent integration** (N-CDI): We generalized the CDI task by considering $N$ noisy input streams and $N$ static context inputs, which we call the N-CDI task. Only one of the $N$ static context inputs was 1, indicating which input should be integrated to compute the output. Each noisy input stream was generated the same way as described for the CDI task. As in the CDI task, the RNN was trained to output zero at $t = 0$, and $\pm 1$ at $t = 750$ ms depending on the sign of $\mu$ for the relevant input. We trained networks from $N = 2$ to $N = 6$ contexts.

## 2.3 Learning Rules

We trained RNNs with four different learning rules: stochastic gradient descent with gradients computed using backpropagation-through-time (BPTT), a genetic learning rule (GA) [21], a Hebbian (H) learning algorithm [18], and the Full Force (FF) algorithm [20]. For all learning rules, training was terminated when accuracy on a validation dataset of 2,000 trials exceeded 90%.

**BPTT**: Gradients were computed using backpropagation through time over the entire trial length. After computing gradients, we updated parameters using the Adam optimizer with a mini-batch size of 500 trials and a learning rate of 0.0005.

**GA**: Each iteration of the GA involves producing a more accurate generation of RNNs from the prior generation. We started with an initial population of 50 RNNs with random initializations as described in Section 2.1, termed generation zero. We then computed the loss of generation zero RNNs over 500 trials. The five RNNs with the lowest loss were then used to produce the next generation (generation one). The next generation of RNNs was constructed by randomly choosing one of the five best RNNs from the prior generation and applying a small amount of noise to all the weight matrices. The noise was drawn from $\mathcal{N}(0, 0.005)$. This resulted in 50 new RNNs. The original 5 RNNs (without perturbation) were also included in the next generation. This process was repeated until our validation criterion was met. The final RNN was the RNN with the lowest MSE in the final generation.

**H**: We used the learning rule described in Miconi [18]. In brief, during training trials, all weights, accumulated an eligibility potential. The weight $w_{i,j}$, which is the $(i, j)$ element of $\mathbf{W}_{\text{rec}}$, had an eligibility potential $e_{i,j}$, given by,

$$e_{i,j}(t) = e_{i,j}(t-1) + \left[ r_j(t-1) \times (x_i(t) - \bar{x}_i) \right]^3 \tag{3}$$

where $r_j$ is the activation of artificial unit $j$ and $x_i$ is the pre-activation value of artificial unit $i$. $\bar{x}_i$ designated the average pre-activation value of artificial unit $i$ across the trial. At the end of each trial, the weights were updated according to the eligibilities and the discrepancy between the current trial loss, $R$, and a running average of past losses for this trial type, $\bar{R}$,

$$\nabla w_{i,j} = \eta e_{i,j} (R - \bar{R}) \tag{4}$$

for a learning rate, $\eta = 0.01$. This learning rule therefore alters synapses (weights) when the activity of post-synaptic and pre-synaptic neurons are correlated during training, mimicking Hebbian plasticity. In this learning rule, the RNN output is a readout of a single unit [18]. These Hebbian networks used $f(\cdot) = \tanh(\cdot)$, since the shifted $1 + \tanh(\cdot)$ resulted in training difficulties.

**FF**: We used the learning rule proposed by DePasquale et al. [20]. In short, this learning rule uses a recursive least squares algorithm to minimize the difference between the RNN's output and the

target output. Additionally, recurrent activations are constrained to match the activations of a second randomly initialized RNN that receives both task inputs and outputs. These FF networks used $f(\cdot) = \tanh(\cdot)$, since the shifted $1 + \tanh(\cdot)$ resulted in training difficulties.

## 2.4 Finding RNN attractor states

To study the dynamical mechanisms used by RNNs, we computed their attractor states. An attractor state is an activity state $\mathbf{x}^* \in \mathbb{R}^{50}$ where $\mathbf{x}_{t+1} = \mathbf{x}_t$. We therefore solved for attractor states by optimizing over $\mathbf{x}^*$ satisfying Equation 1

$$\mathbf{x}^* = (1 - \alpha)\mathbf{x}^* + \alpha \left( \mathbf{W}_{\text{rec}} f(\mathbf{x}^*) + \mathbf{W}_{\text{in}} \mathbf{u}_t \right) \tag{5}$$

Because $\mathbf{x}^*$ depends on the input $\mathbf{u}_t$ to the RNN at time $t$, we set the input to be a static value when solving for attractor states. We solved for these fixed points by using a numerical solver, based on the Levenberg-Marquardt algorithm [24]. Due to numerical precision, we termed any states satisfying $\|\mathbf{x}_{t+1} - \mathbf{x}_t\| < \varepsilon$ to be an attractor, where $\varepsilon = 10^{-8}$. In some analyses, we also computed slow points along a line attractor, which corresponded to states with $\varepsilon = 1$. After finding fixed points, we clustered them based on their topology, described further in Section 2.6.

## 2.5 SVCCA to assess similarity of RNN activity

We quantified how similar the representational geometry, that is, the activity of two different RNNs, was to the same inputs using singular value canonical correlation analysis (SVCCA) as described in Raghu et al. [25]. SVCCA was also used by Maheswaranathan, Williams, et al. [14] to quantify differences and similarities in representational geometry between RNNs. To compute the SVCCA, we provided multiple test inputs to the RNN. RNN activity was then projected onto its top $K$ principal components (PCs), with $K$ chosen so that the PCs explained over 95% of the variance. Canonical correlation analysis (CCA) was then performed on the projected activity to determine a correlation between the PC-embedded activations of two RNNs. The SVCCA was the maximum correlation between the two projected representations subject to a linear transformation.

## 2.6 Multi-dimensional scaling to visualize attractor topology and representational geometry

We compared an embedding of attractor topologies across RNNs in a similar manner as [14]. For additional details, refer to the Appendix. To compare representational geometry across RNNs, we computed a pairwise distance matrix between all RNNs using SVCCA. Each element of the distance matrix was 1 minus the SVCCA correlation between pairwise RNNs. We used MDS to qualitatively visualize how clustered this pairwise distance matrix was for different tasks. To quantify clustering, we computed the Silhouette cluster score for the pairwise distance matrix using the learning rule as the cluster label. All Silhouette cluster scores were computed on the high-dimensional data, not the MDS projections. MDS projections were used solely for visualization purposes.

# 3 Results

We numerically investigated the impact that learning rule, task complexity, and input noise have on RNN representations and dynamics during perceptual decision-making tasks. We make the following contributions. First, we show that across four learning rules, RNNs adopt a similar attractor topology regardless of the learning rule used. Second, we show that as task complexity increases, the representational geometry of RNNs becomes more individualistic (unique) across different learning algorithms while attractor topologies remain universal. Third, we show that task input noise, which affects task difficulty, can result in different RNN integration dynamics. Fourth, we show evidence that as tasks become more complex, task input noise has a smaller effect on RNN attractor topologies. These results suggest that RNN attractor topologies, and therefore dynamical mechanisms, are similar across learning rules and tasks despite differences in representations.

## 3.1 RNN attractor topologies are universal across learning rules

We trained RNNs to perform the RDM task and the CDI tasks using the previously described learning rules. All RNNs successfully performed the tasks with greater than 90% accuracy following training

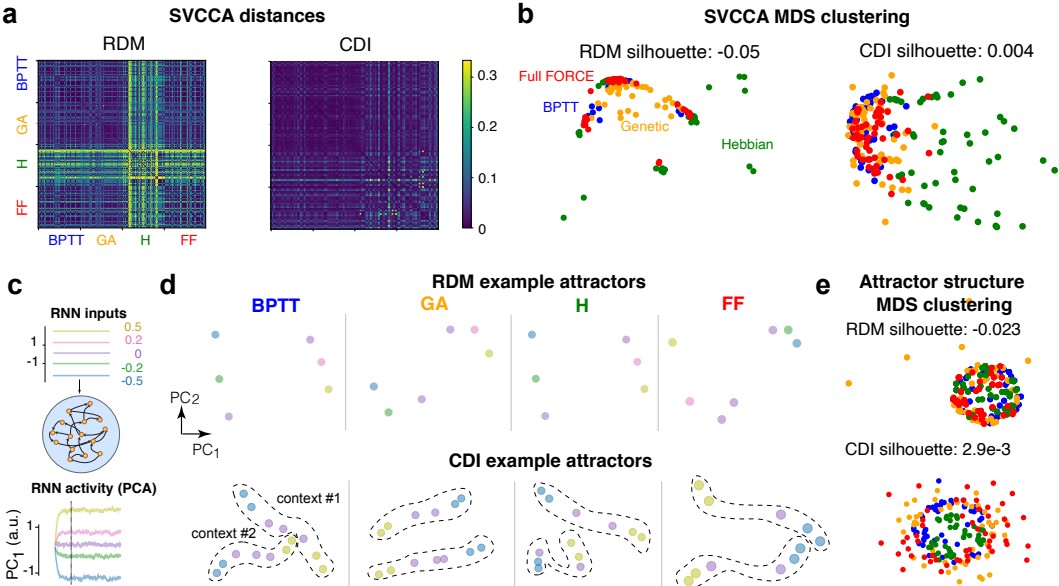

Figure 1: **Representation and attractor structure across learning rules.** **a**) SVCCA distance matrices for RDM (left) and CDI (right). There are 50 trained networks for each learning rule. BPTT, GA, and FF exhibit similar SVCCA distances, but H is different. (**b**) SVCCA MDS clustering shows that BPTT, GA, and FF networks are highly overlapping, while H representations are more dispersed. Each dot corresponds to one RNN. **c**) Attractor calculation intuition. Different inputs to the network (top) lead to different RNN equilibrium representations (bottom, which shows stable activity along the first principal component). We show 5 different attractor states corresponding to inputs: $\{-0.5, -0.2, 0, 0.2, 0.5\}$. **d**) Example attractors for the 5 different input levels for a single RNN trained with all four learning rules for RDM (top) and CDI (bottom). Attractors are visualized in the top two principal components. **e**). Clustering across RNNs trained for BPTT, GA, H, and FF (50 networks each). Networks are highly clustered in attractor structure.

with any of the four learning rules. We assessed the similarity of RNN representations by computing a pairwise distance matrix between all RNNs based on their SVCCA score, shown in Figure 1a. We used MDS clustering to visualize RNNs based on similarity of neural representations, revealing that network representations were largely overlapping following training with BPTT, GA, and FF learning rules (Figure 1b). Interestingly, Hebbian RNNs exhibited less overlap with RNNs trained with other learning rules, which may be because Hebbian RNNs use an arbitrary recurrent neuron as the output [18]. We therefore performed these analyses after training non-Hebbian RNNs to use a single neuron readout, and observed consistent results (Appendix Figure 11). Overall, this suggests that representations in RNNs are similar across BPTT, GA, and FF for the RDM and CDI tasks.

We analyzed the attractor states of all trained RNNs to investigate their dynamical mechanisms and whether they differed across learning rules. We solved for attractor states of each network under five different probe input conditions: $\mathbf{u} = \{0.5, 0.2, 0, -0.2, -0.5\}$ (Figure 1c). We visualized these attractor states in the principal components (PCs) of the activity, with example networks for each learning rule shown in Figure 1d. In the RDM task, non-zero inputs resulted in a single attractor state. We observed that at zero input, the network had multiple attractor states, which we elaborate on in Section 3.3. In the CDI task, RNNs instantiated relatively similar attractor topologies, with a set of input-based attractors for each context, previously shown in Mante et al. [1] and Maheswaranathan et al. [14]. All RNNs also exhibited a line attractor of slow points, discussed further in Section 3.3.

Visualization in the PCs is limited, since these attractors are a property of the high-dimensional RNN activity space. We therefore computed attractor structure similarity in the space of the RNN activity. We clustered RNN attractors associated with static inputs near zero based on the topology of their nearest neighbors and used MDS to visualize the resultant clusters (see Appendix and [14]). These results are shown in Figure 1e. We found attractor topologies were indistinguishable across RNNs trained with different learning rules. Silhouette cluster scores were close to zero across all learning rules. Our results contribute to literature finding evidence that there is universality and invariance of

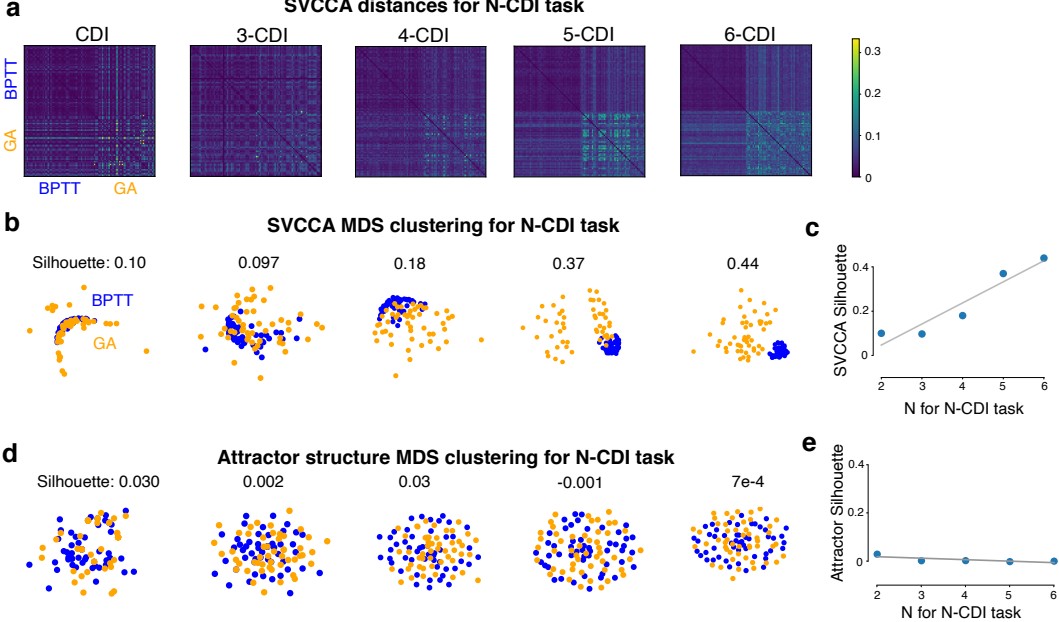

Figure 2: **Effect of task complexity on representational geometry and attractor topology. (a)** SVCCA distance matrices for N-CDI task varying $N = 2$ (left) to $N = 6$ (right) comparing BPTT and GA networks. As the task became more complex, block structure was observed, indicating larger distances between BPTT and GA representations. **(b)** Visualization of SVCCA MDS clusters for $N = 2$ (left) to $N = 6$ (right). As $N$ increases, BPTT and GA RNNs representations cluster. **(c)** SVCCA cluster silhouette score increased as the number of contexts in N-CDI increased. **(d)** Visualization of attractor MDS clusters. As $N$ increases, BPTT and GA attractor structures remain overlapping. **(e)** Attractor topology silhouette score did not show a trend with the number of contexts.

attractor structure across different RNN architectures and activation functions [14]. In particular, we show that different learning rules exhibit universal attractor structure.

## 3.2 Increasing task complexity results in more individualistic representations, but not dynamics

To assess the degree to which representations and attractor topologies varied with task complexity, we performed the N-CDI task, varying the number of contexts and inputs in the task. We speculated that increasing task complexity may affect neural representations following training with different learning rules since more task-related information would need to be stored in the network. Concurrently, increasing task complexity likely reduces the space of task solutions, constraining possible solutions. We found that, as tasks became more complex, H and FF training did not meet the accuracy termination criterion when $N > 2$, that is, these learning rules did not lead to high-performing networks. We therefore report results comparing BPTT and GA.

BPTT and GA trained RNNs had larger MDS distances (Figure 2a) as the number of CDI contexts increased from $N = 2$ to $N = 6$, indicating different representations. As a result, we observed stronger SVCCA MDS clustering, shown in Figure 2b. We quantified this by measuring the Silhouette scores of the BPTT and GA SVCCA clusters, finding they generally increased with increasing $N$, as shown in Figure 2c. These results suggest that increasing task complexity leads to representations that form distinctive clusters depending on the choice of learning rule. That is, RNN representations became more individualistic as task complexity increased since trained RNNs had distinct (individual) representations instead of the universally shared representations. This is important to consider, especially as computational neuroscience studies typically compare RNN representations to neurophysiological representations. In contrast, we found that across all values of $N$, BPTT and GA RNNs shared a universal attractor structure (Figure 2d,e). These results suggest that even as representations become more individualistic for complex tasks, attractor topologies may remain universal across learning rules.

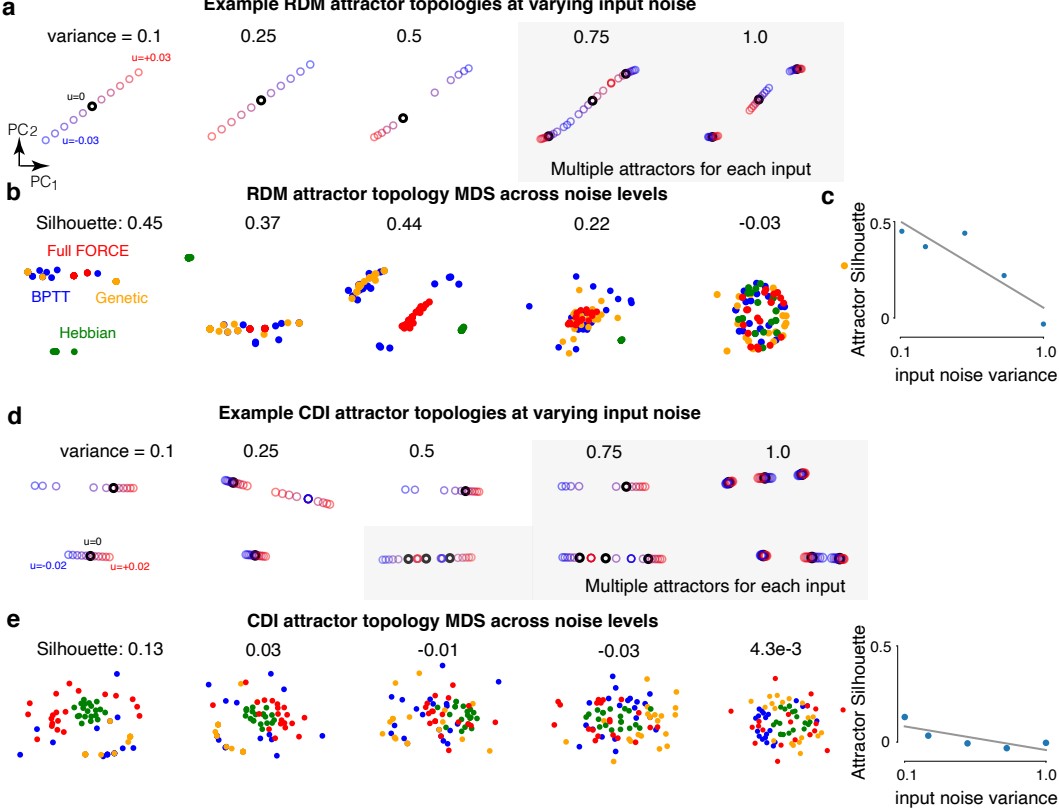

Figure 3: **Effect of input noise on attractor structure. (a)** Example RNN attractor structure at different input noise variance. When the noise variance is sufficiently large, the RNN instantiates multiple attractors for low inputs. **(b)** As the noise increases, there is more universality in attractor structure across learning rules. **(c)** Attractor structure clustering decreases with increasing input noise. **(d-f)** Same as (a-c) but for the CDI task.

## 3.3 Task input noise can alter RNN integration strategies

Because RNNs are used to propose candidate mechanisms for neural computation, it is appealing that RNNs exhibit similar dynamics across different training approaches and task complexities, suggesting robustness to particular hyperparameter settings, network architectures (as detailed in [14]) and learning rule. However, this led us to ask: under what settings, if any, might a network exhibit distinct dynamics? We reasoned that an important consideration is task input design, which is determined by the experimenter. For example, in the CDI task, Mante et al. [1] chose the standard deviation of the input noise to be $31.623\sqrt{\Delta t}$, where $\Delta t$ is the time step of the Euler update for the RNN, resulting in a noise variance of 1. But Miconi [18], also modeling the CDI task, set the noise to 0.5 and the mean to 1 (instead of at most 0.1875), leading to a difference in task difficulty. However, the input noise may dramatically affect the integration strategies: tasks with high input noise may employ more robust integration strategies. We ultimately find that, depending on the input noise, RNNs may employ different integration mechanisms, including for the CDI task [1].

We trained RNNs to perform the RDM and CDI tasks under varying input noise levels with the four learning rules. As input noise increased, there was a greater multiplicity of attractors for small input values (shaded attractor topologies in Figure 3a,d). Whereas low input noise levels instantiated one attractor for each input, increasing levels of noise resulted in the emergence of multiple attractors for small inputs. Attractors associated with small input values are of particular interest since they correspond to the networks state in the absence of any meaningful input. Examples of these attractor topologies are visualized in Figure 3a,d, where representative attractor topologies are shown for BPTT. Interestingly, the topological structure of attractors was invariant across learning rules for the high noise tasks. However, following training on the lower noise task variants, attractor topologies were

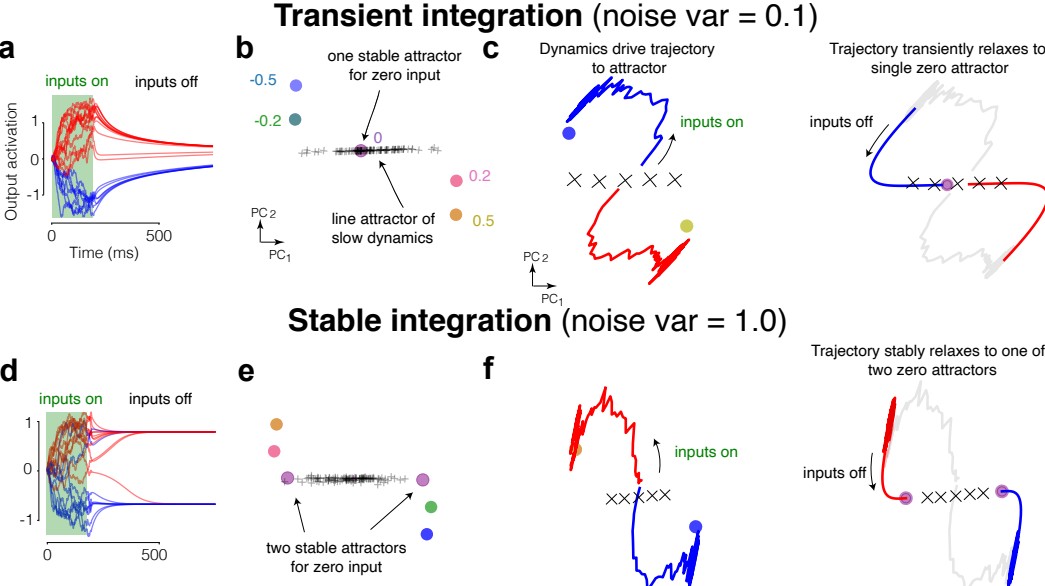

Figure 4: **Two different input strategies based on input noise.** Panels (a-c) illustrate transient integration, while (d-f) is for stable integration. **(a)** RNN output activation for positive (red) and negative (blue) inputs. After inputs turn off, the RNN output slowly decays. **(b)** Attractor topology for a transient RNN. Dots are attractors. Numbers next to them correspond to the setting of the input, $\mathbf{u}_t$. There was a single attractor for $\mathbf{u}_t = 0$. There was also a line attractor of slow dynamics under zero input, denoted by gray x's. **(c)** When the inputs turn on, the RNN state is driven towards attractors (blue, yellow dots). When the input turns off, the trajectories relax to a single attractor corresponding to zero input. This decay is relatively slow along a line attractor of slow dynamics. **(d)** RNN output activation for a RNN with stable integration. After the inputs turn off, the RNN output maintains a stable value. **(e)** Attractor topology for stable RNN. Now, at an input of zero (purple), there are two attractors. **(f)** After the dots turn off, the trajectory stably relaxes to one of the two attractors corresponding to zero input, holding memory of the input stably.

not universal across learning rules (Figure 3b,c). The effects of input noise and task complexity on attractor topologies is further discussed in Section 3.4. Together, these results suggest that task noise affects attractor topology, and when the task is too easy (low noise), learning rules may converge to different dynamical mechanisms.

We found that this bifurcation from single to multiple attractors results in two fundamentally different modes of integration, which we term "transient" and "stable" integration for the single and multiple attractor structures, respectively. We interrogated these dynamical mechanisms by providing a pulse of input evidence to the RNN. In these experiments, the inputs to the network are turned off (set to zero) midway through the trial (Figure 4a,d, "inputs off"). Single attractor networks employed a transient integration mechanism, which is a form of leaky integration, causing the output to slowly forget its integrated state (Figure 4a). When the input was pulsed on, the RNN state evolved to the single attractor corresponding to the input level (Figure 4c, left). However, when the input was turned off, the network instantiated a single attractor at the zero input level, which caused the network state to slowly relax to an attractor corresponding to zero output (Figure 4c, right). The network, under zero input, also, instantiated a line attractor with slow dynamics (Figure 4b,c), consistent with the dynamics reported in Mante, Sussillo, et al. [1]. The network therefore decayed to the single attractor at zero, albeit slowly along the line attractor. This corresponded to a slowly decaying output value, reflecting leaky integration of the input.

In contrast, the multiple attractor networks employed a stable integration mechanism. In particular, small inputs resulted in multiple fixed points, shown in Figure 4e. When turning off the input, the RNN state converged to a nearby attractor that maintains a stable memory of the sign of the pulsed input (Figure 4f). This enabled the RNN to stably and indefinitely output a non-zero value (Figure 4d), corresponding to the sign of the prior integrated evidence, in contrast to the leaky integration mechanism. In both mechanisms, there is a line attractor of slow dynamics that the RNN

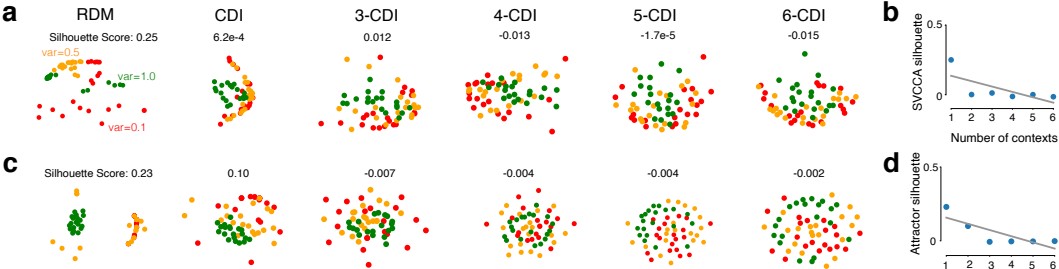

Figure 5: **Input noise and task complexity.** We show SVCCA and attractor topology clustering for BPTT networks trained at input noise variances of 0.1, 0.5, and 1.0. **(a, b)** As task complexity increases (more contexts), representations are highly overlapping. **(c, d)** Increasing task complexity results in similar attractor topologies across different input noise variances.

relaxes along. However the key difference between these mechanisms is the presence of multiple attractors at low inputs. The same analysis for the RDM task in Figure 4 is also shown for the CDI task in the Appendix. Together, these results show that task design, in particular, input variance, can have a stronger effect on the dynamical mechanism than learning rule.

### 3.4 Attractor topologies are robust to input noise as tasks become more complex

We previously found that, as tasks become more complex, networks trained with different learning rules maintain similar attractor topologies. This may reflect that as tasks become more complex, this constrains the space of potential solutions. Additionally, we found that RNNs trained at different levels of input noise instantiated different attractor topologies and therefore dynamical mechanisms. We therefore investigated the effect of input noise on attractor topology in the setting of increasingly complex tasks.

We trained BPTT networks at three different noise levels: variances of 0.1, 0.5, and 1.0. For the RDM task, RNNs trained at different levels of input noise instantiated distinct attractor topologies. For the 2-CDI task, however, attractor topologies were less strongly clustered by input noise and appeared more universal. We subsequently trained the N-CDI task for these three input noise levels up to $N = 6$. We found that as $N$ increased beyond 3, attractor topologies were indistinguishable across all three noise levels. Figure 5 depicts attractor topologies following training at different levels of input noise for all tasks. These results suggest that although input noise may affect RNN dynamics, these effects on attractor topology diminish for sufficiently complex tasks.

## 4 Discussion

We studied the effects of learning rule, input noise, and task complexity on RNN dynamics. We found that an RNN's attractor topologies are invariant to the choice of learning rule across different decision-making tasks. We also observed this conclusion in a simple working memory task (Appendix). We found the amount of noise present in the task during training can change an RNN's dynamics when the task is sufficiently simple. These results suggest that when a task is simple, hyperparameters may affect RNN dynamics. Consistent with these results, we also found that activation function affects attractor topology for RDM, but not the more complex CDI task (see Appendix).

We characterized two distinct dynamical mechanisms used for integrating the inputs to an RNN: a transient and stable integration mechanism. In the case of high input noise, we find a dynamical mechanism consistent with that reported by [1] following training in the CDI task. Alternatively, training with low input noise led to a mechanism with transient integration. While the topological structure of attractors did not vary, the representational geometry of RNNs was different following training with different learning rules on more complex N-CDI tasks. Our findings here are consistent with those by Maheswaranathan, Williams, et al. [14] that dynamics and computational scaffold are invariant over various RNN design choices while representational geometries are not.

These results contribute to laying a foundation for interpretability in using RNNs to study computational neuroscience. We focused here on traditional RNNs on relatively simple neuroscience tasks to

better understand the effect of learning rules on attractor topologies and representational geometries. Our main contribution is showing that dynamical mechanism, as assessed through attractor structure, remains conserved across different learning rules and task designs for sufficiently complex tasks, even if representational geometry is not. This is an important finding since many existing computational neuroscience studies directly compare representations from artificial neural networks to neural recordings. While our results suggest that for tasks as complex as N-CDI, we observe universality in attractor topology, future work should extend these results to more complex tasks. More broadly, having a framework that allows better understanding of the validity of neurobiological inferences from modern neural networks during realistic tasks is important for building a foundation for RNN use in neuroscience studies.

## Acknowledgments

We thank Niru Maheswaranathan for helpful discussions on the attractor topology comparison. BM was supported by the NIH NIGMS training grant GM008042. MK was supported by the National Sciences and Engineering Research Council (NSERC). JCK was supported by NSF CAREER 1943467, NIH DP2NS122037, the Hellman Foundation, and a UCLA Computational Medicine AWS grant. We gratefully acknowledge the support of NVIDIA Corporation with the donation of the Titan Xp GPU used for this research.

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
