# OpenReview forum: "Learning rule influences recurrent network representations but not attractor structure in decision-making tasks"
_NeurIPS.cc/2021/Conference — NeurIPS 2021 Poster_

### Official Review · Reviewer_GYy3 · 2021-07-13

**Rating:** 5
**Confidence:** 5

**Summary:**

The authors extend the recent work of Maheswaranathan et al to explore how learning rule and task complexity affect the solutions obtained by recurrent neural networks trained on neuroscience-inspired tasks.
Specifically, four different learning rules are used to train networks on a suite of similar tasks with systematically increasing complexity. The authors show that learning rule has some effect on representation geometry, and this effect grows with task complexity. Attractor topology, however, remains universal for almost all tasks and learning rules.
The authors also show some instances in which network dynamics are influenced by input noise. This sensitivity, however, decreased as task complexity increased.


**Limitations And Societal Impact:**

Societal impact – not relevant.
Limitations: mentioned above.


**Main Review:**

The question of individuality and universality of solutions obtained by trained recurrent neural networks is an important and timely one. These networks are used to compare with experimental data, and it is important to understand how the nature of solution is affected by various design choices.
The specific extensions studied here – learning rule and task complexity – are widely used to optimize networks, so it is critical to understand how each criterion biases the obtained solutions, especially when claiming similarity between RNNs and brains. Of particular note is the use of genetic algorithms, that are not often used in this context, and could hence provide novel viewpoints on the space of solutions. Furthermore, the systematic approach to task complexity is a promising approach to understand its effect.
Despite these strengths, there are several major issues with the current submission: The novel insights beyond the work of Maheswaranathan et al are not large; The effects uncovered by the authors are not explored; The dynamical system analysis is non-standard and not justified appropriately.
1.	As the authors state, this work is very much in line with that of Maheswaranathan et al.  As such, the added value should be much more clearly stated. In particular, the reasons for discrepancy should be studied. The authors speculate that “increasing task complexity… constraining possible solutions”. And yet the empirical finding is that solutions of complex tasks are more individualistic. This seems like a promising avenue for exploration – but it is not pursued.
2.	Hebbian learning is distinct from all other rules. This is interesting. Why is it so? The topic is not pursued.
3.	The analysis used to arrive at the conclusion is composed of many processing steps. At the end of all these steps, we are left with a single number (Silhouette score) that describes variability. Compressing a complex phenomenon like the solution found by a network to a single number is desirable in order to produce figures like 2C and 2E. But it also precludes an understanding of what the underlying reasons for these observations are. It would be very helpful to accompany such analyses with several intermediates like comparing PCA (or other dimensionality reduction methods) of various networks.
4.	The authors define fixed points as having a speed of 1e-8 and below, and slow points of a speed of 1 and below. This is a very high threshold for the slow points (compare to Sussillo & Barak 2013). Furthermore, the analysis used inputs that are close to zero to define some ordering between fixed points. If I understand correctly, the authors actually find slow points that have a speed considerably lower than 1. By slightly perturbing the input from zero, these slow points cross the threshold of being considered fixed points. First – this is my guess. Second – this could be an interesting heuristic for reverse engineering trained networks. Third – none of this is explained or justified in the appendix. Standard analysis requires using zero input. Other choices might be useful, but should be justified and clearly stated.

Specific points:
5.	The authors analyze one task with varying complexity and find that the dynamics are almost invariant. It could be that other tasks have more variability, and hence could be better suited to answer the main question posed here – does learning rule affect solution?
6.	Silhouette score is used extensively. It could be useful to provide some intuition on the meaning of the various values.
7.	Figure 1 caption “networks are highly structured in attractor structure”. This sentence is somewhat confusing when combined with a low silhouette score
8.	Figure 2B 5-CDI panel shows two clusters for the GA. Why is that? Such observations are not mentioned or analyzed in the paper.
9.	Figure 3A the attractor changes shape from a line to a sigmoid. Why is that?
10.	Figure 3a: It is unclear where the slow areas appear in relationship to the attractors, which differences arise when the Silhouette score is high (e.g. figure 3B for low variance) and what is the contribution of the learning rule to this variability. This is especially needed because the Hebbian learning rule appears to dominate the variability.
11.	L271: “memory of the pulsed input”. Should this be memory of the sign of the input?
12.	Figure 4. Perhaps looking at the actual speed values of points along the attractor can be informative here.
13.	Figure 5 caption is hard to understand
14.	L302: “two distinct dynamical mechanisms used for integrating the inputs”. I’m not sure whether it was demonstrated how these mechanisms are used for the task.
15.	Figure S6 in appendix. The panel D, which represents stable integration looks quite similar to the  transient panel A. This is not commented upon.
16.	In general, interpretation and explanation of the numerical results would greatly strengthen the paper.


**Time Spent Reviewing:**

3

---

> ### Author Response · Authors · 2021-08-10
> **Response**
>
> Thank you for your comments and suggestions on this paper.
>
> To your point on the value added on top of Maheswaranathan et al. and a potential discrepancy:
> >First, we wanted to clarify that the potential discrepancy mentioned is not a discrepancy, and we will clarify the text to resolve this misunderstanding. Our statement that "increasing task complexity... constraining possible solutions, "refers to constraining the networks dynamics, which describe the RNN’s computation and therefore mechanism (for a nice review, see [1] "Computation Through Neural Population Dynamics"). The proxy for dynamical mechanism is attractor topology, not network representation.  For example, we would consider networks that do selective integration via line attractors (as observed in our N-CDI task, and further described in [2]) to employ the same mechanism, even though the particular representational geometry of the activations along the line attractor may differ.  While representations can become more individualistic as task complexity increases, we observe that the attractor structure that defines its dynamical mechanism is highly overlapping as complexity increases (Fig5c,d, and Appendix Figure 7, where for simpler tasks, input noise and activation function strongly affects attractor topology, but not when the task becomes more complex).  This is indeed consistent with Maheswarnathan et al.  [3], who also found that representational geometry exhibited differences between RNN architectures and activation functions, but universality in attractor structure. There is therefore no discrepancy here. We will clarify this explanation in our Discussion.
>
> >We wish to highlight that there are significant conceptual advances made over the study by Maheswaranathanet al. [3]. We make the following contributions that extend beyond their initial work:
> * We looked at how the learning algorithm affects the RNN’s dynamics, something not explored by [3] which used BPTT for all architectures.
> * We used the body of N-CDI tasks to answer questions about how task complexity affects representation and attractor similarity.
> * We found a design variable, input noise, that strongly affects attractor structure, something not explored by [3]. Further, we showed how this could result in different integration mechanisms (Figure 4), which has significance for neurophysiological experiments and hypotheses (e.g., continuous vs discrete integration). This also shows that convergence in attractor topology is not due to a trivial lack of alternative solutions.
> * We found that task noise is less of a concern when the task is sufficiently complex.
>
> >We  will  highlight  these  conceptual  advances  over  Maheswarnathan  and  colleagues  more  clearly  in  the discussion, so it can be more easily understood how our work significantly advances prior art.
>
> To your point on Hebbian learning being distinct:
> >Our initial reason for training a Hebbian network with a single unit readout was to replicate the training rule exactly as in the literature [4], as this is how they are typically used in computational neuroscience studies.  We agree it is important to control for this difference.  We have therefore since trained additional models using the BPTT and GA learning rules using a single recurrent unit as output to perform the RDM and CDI tasks. With this control, we find that the representational geometries appear more similar across learning rules and attractor topologies remain consistent with Figure 1. We will include this as an Appendix Figure.
>
> To your point on incorporating intermediate comparisons like PCA:
> >Thank your for raising these points, which we agree with.  We will include additional figures showing intermediate PCs, fixed point structures, and computational steps in the Appendix Figures. We believe this will provide additional insight and allow readers to make more detailed comparisons. To your point regarding the observation that our threshold for slow points was too large: We actually choose the threshold of the slow point to exactly match the threshold used by Mante, Sussillo et al., Nature 2013[2] for the CDI task, who used a threshold of 1.0 to find slow points (see page 15 of their Supplementary Methods). Furthermore, we did explore this threshold and note that making the threshold more stringent (1e-4) had little effect on the slow points uncovered and did not alter any conclusions. This value is therefore entirely consistent with the primary study on these tasks. (We note that we used the same task hyperparameters as in [2].)
>
> To your point regarding fixed points found by perturbing the input from zero:
> >Thank you for bringing this up, and the opportunity to clarify and justify our fixed point analysis.  We wish to clarify that we used standard practices for dynamical and fixed point analyses, consistent with prior work, and we will more clearly describe this in the Appendix. The standard for RNN studies is to choose inputs to find fixed points that make sense for the particular task, and frequently, this corresponds to inputs that are not zero. (As an aside, identifying fixed points commonly do not use zero inputs for decision-making and motor tasks, e.g., see [2] and [5]; and they are often zero for tasks that require memory, since these networks must remember prior inputs even when they are no longer present and the input is zero, such as in the 3-bit flip flop task of Sussillo and Barak 2013 [6].) For example, in the CDI task, Mante et al. found stable fixed points corresponding to a non-zero context input [2]. Under zero context input, there would be no stable fixed attractor corresponding to a correct decision since there would be no context to adhere to. These stable attractors only appear when the contextual input is turned on (which is why [2]says the motion and color attractors "never exist together"), since these inputs define new equilibrium states for a particular context, with distinct dynamics (see Eq 1 of our paper). It is precisely the different context inputs that instantiate different stable attractors and line attractors, leading to the selective integration described in [2]. For other examples of studies with neuroscientific insight from input-dependent fixed points, see [5] and [7]. One more example is that Maheswaranathan and colleagues used different inputs to identify attractor structure for their sinewave task [3].
>
> >The reason we change the input in this study is because we want to understand how the RNN’s equilibrium state changes in different contexts of the N-CDI task, and under difficult (low coherence) and easy (high coherence)trials. Our justification for considering the fixed points in a small neighborhood of input values near zero is because these networks must integrate a noisy input signal that is close to zero, and we want to understand how the network state converges in these difficult settings. Therefore, the dynamical landscape the network instantiates not just at zero but in a neighborhood of input values around zero will influence the computational mechanism and behavior of the network on the task. RNNs that instantiate multiple attractors over a larger neighborhood of inputs near zero input will exhibit different dynamics (stable integration) than networks that instantiate only a single attractive fixed point for each input about zero (transient integration), as we document in Figure 4. We believe this is an important result of the paper. (It is worth noting that multiple attractors also emerge for zero input when task input noise is sufficiently large).
>
> >We also note that our computation of the slow points is over zero coherence input, also consistent with Mante et al. [2], as in this case, our goal is to not find the equilibrium states of the network at a given input, but to observe the slow points of the dynamical landscape over which this integration occurs. In this case, using zero coherence input was the correct choice. Finally, we wish to clarify that we do not characterize any states transitioning from being slow points to fixed points. We merely solve for the slow points and the fixed points using different thresholds. We appreciate the nuance in these details is important to get right and will carefully describe these details to the Appendix so it is clear that these choices are standard and justified, as well as references to prior work that use this approach. Thank you for bringing this to our attention.
>
> To your specific points:
> >Thank you for your careful read of the manuscript in raising these specific points. We adjusted places where you asked for further textual explanations or figure clarifications / additions (7, 11-14).
>
> For other specific points:
> * 5.  Regarding tasks:  we now include a delay non-match to sample (DNMS) [4] working memory task that upholds our conclusions, which will be added as an Appendix Figure.
> * 6.  We will update the text to include an intuition of the silhouette score.
> * 8. We will clarify the variance amongst GA networks is due to its random seed and the overall variance in convergence of GA representations.
> * 9. This could be due to several factors, including that with more attractors, they are less likely to lie on a linear manifold, which is found via PCA.
> * 10. Regarding slow points not visualized in Fig 3a, we will point the reader to Fig 4b and 4e, which shows the slow regions (line attractor) with respect to the fixed points. Indeed, the largest variations come from Hebbian, which according to our prior control, is related to its output readout.
> * 11. We agree this language is more conservative and will update it as suggested.
> * 15.  While they look similar, Fig S6D stabilizes to a stable value while Fig S6A does not.  We will clarify this in the legend.
> * 16. Thanks for this feedback, we will aim to improve explanation and interpretation through textual changes and additions.

---

> > ### Author Response · Authors · 2021-08-10
> > **References**
> >
> > 1. Vyas S, Golub MD, Sussillo D, Shenoy KV. Computation Through Neural Population Dynamics. Annu Rev Neurosci. 2020;43: 249–275.
> > 2. Mante V, Sussillo D, Shenoy KV, Newsome WT. Context-dependent computation by recurrent dynamics in prefrontal cortex. Nature. 2013;503: 78–84.
> > 3. Maheswaranathan N, Williams A, Golub M, Ganguli S, Sussillo D. Universality and individuality in neural dynamics across large populations of recurrent networks. In: Wallach H, Larochelle H, Beygelzimer A, d’Alché-Buc F, Fox E, Garnett R, editors. Advances in Neural Information Processing Systems 32. Curran Associates, Inc.;2019. pp. 15603–15615.
> > 4. Miconi T. Biologically plausible learning in recurrent neural networks reproduces neural dynamics observed during cognitive tasks. Elife. 2017;6. doi:10.7554/eLife.20899
> > 5.  Kao JC. Considerations in using recurrent neural networks to probe neural dynamics.  J Neurophysiol.2019;122: 2504–2521.
> > 6. Sussillo D, Barak O. Opening the black box: low-dimensional dynamics in high-dimensional recurrent neural networks. Neural Comput. 2013;25: 626–649.
> > 7. Chaisangmongkon W, Swaminathan SK, Freedman DJ, Wang XJ. Computing by robust transience: how the fronto-parietal network performs sequential, category-based decisions. Neuron. 2017;93: 1504–1517.e4.

---

> > > ### Comment · Reviewer_GYy3 · 2021-08-31
> > > **Keeping my score**
> > >
> > > I thank the authors for the detailed responses.
> > >
> > > After carefully reading all the reviews and responses, I still feel this paper needs more work before publication.
> > >
> > > The main novel result is that there are two different dynamical landscapes that can solve this task: transient and stable. It is not clear if these matter with regards to how the network actually solves the task. It seems that the integration phase is identical, and only the relaxation (after the task is complete) is different. This is still a valid and interesting result, but underexplored. For instance - does the silhouette scores fully capture this effect? What is the fraction of stable/transient networks in each architecture / learning rule / noise level? This is a very natural and simple analysis to do, but is missing.
> > >
> > > My overall impression is that the paper suggests interesting directions, but only presents preliminary results.

---

> > > > ### Author Response · Authors · 2021-09-07
> > > > **A few clarifications**
> > > >
> > > > We want to clarify that the existence of two different dynamical landscapes (figure 4) is just one of our multiple novel results. Additionally, we show that attractor topologies are conserved across various learning rules for the RDM and CDI task (figure 1). To our knowledge, this has not been previously explored. We also revealed that dynamical mechanisms are more likely to be conserved for more complex tasks across different learning rules (figure 3), activation functions (appendix figure 6), and task design choices (figure 5). This relationship between dynamics and task complexity has also, to our knowledge, not been previously explored.
> > > >
> > > >
> > > > We wish to further clarify that these two different dynamical landscapes are clearly distinct mechanisms, and there is a clear picture as to how the network solves the task. Inputs drive the network state, with positive and negative inputs moving the network state in directions opposite of the output readout (Wout) separatrix, resulting in distinct output readouts. Given your comment, we will add this separatrix to Figure 4 to increase clarity. The network state is an internal representation of the integrated input towards a stable attractor, and is therefore a critical representation. During the integration phase there exists only a single attractor state for the transient mechanism but there exists multiple attractor states for the stable integration mechanism. Therefore, the RNN’s state is traversing a fundamentally different energy landscape during the integration phase depending on which dynamical mechanism is instantiated. In the single attractor landscape, there is a basin of attraction that the network state is drawn towards, but if inputs are turned off, it leaves this basin of attraction slowly. In the multiple attractor landscape, there are multiple basins of attraction, which also imply a barrier between basins. Behaviorally, we explored the differences during the relaxation phase as we believe these are most interesting in a computational neuroscience setting. You are correct in that both these mechanisms result in the RNNs being able to perform the integration based decision making task. In computational neuroscience, RNNs are often trained to infer what computational mechanism the brain may be using. Thus it is possible for RNNs to adopt distinct dynamics during training on simple tasks with different design choices.
> > > >
> > > >
> > > > The Silhouette score is a summary measure of this effect because it quantifies how similar the attractor structure is, but the Silhouette score by itself cannot illustrate all the nuances of these dynamics. This is why we detailed differences in the mechanism further in Figure 4.
> > > >
> > > > We did not train different architectures on this task. You are correct that the analyses asked for are simple to do, and while we discussed the trends in the text, we did not report the numbers in our manuscript. In response to your comment, for noise level, we found that at variance levels of 0.1 and 0.25 0% of BPTT, GA, HEB, and FF RNNs instantiated multiple attractors. For a variance of 0.5 we found that 35% of BPTT and 100% of FF networks instantiated multiple attractors while 0% of GA and HEB networks instantiated multiple attractors. At a variance of 0.75, 90% of BPTT, 100% of GA, 100% of FF and 0% of HEB RNNs instantiated multiple attractors. At a variance of 1.0 we found all RNNs across all learning rules instantiated multiple attractors.We will add this to the Results (space permitting) or else the Appendix.

---

### Official Review · Reviewer_T4eL · 2021-07-15

**Rating:** 8
**Confidence:** 4

**Summary:**

This paper aims to characterize the impact of design choices in RNN-based experiments on learning dynamics and representation geometries, both of which are important data sources when using RNNs to model biological intelligence. It begins by setting up the problem of investigating RNNs’ fitness to model cognitive and neural systems, and drawing a distinction from the most similar work (Maheshwaranathan et al). It then describes a standard continuous-time RNN architecture, three tasks, one of which is a complexity-modular version of another, and four learning rules of which I believe one is local. Along with the model setup, it details evaluation and visualization methods. The bulk of the paper appropriately focuses on results. This goes through the following conclusions: dynamics do not vary with learning rule (implicitly, for more complex tasks), representations vary with task complexity, dynamics do not vary with task complexity, dynamics do vary with learning rule under low input noise with simpler tasks, and finally dynamics are confirmed to not vary with learning rule under low input noise with more complex tasks (suggested explanation being that more complex tasks have a more constrained solution space). The paper ends with a summary of these conclusions and their implications for the use of RNNs in computational neuroscience.

**Limitations And Societal Impact:**

The checklist claims that this work discusses its limitations, but the referenced limitation is “mechanisms in the RNNs should be interpreted as a hypothesis for biological mechanisms”. I wouldn’t consider this a discussion of limitations of this work. I’m not even sure how it’s a limitation at all, but if the idea is that RNNs aren’t a perfect model for biological intelligence and are primarily theorized as one, then that’s a critique of the whole field (in fact, one that this paper is trying to rectify). The one limitation that this work does discuss in the main text is that experiments should extend to more complex tasks, but this is given half a sentence and is posed as a future work. I totally agree and think the paper is complete enough to leave that for future work, but that’s not a sufficient discussion of limitations as per the NeurIPS checklist guidelines. I think the paper should discuss the limited space of tasks (even in the range of complexity of RDM and (N-)CDI) and learning rules, or biological implausibility of some of the learning rules. It’s certainly possible that I misinterpreted the note in the checklist, but I don’t think this paper really discusses the limitations. I also don’t think it would be hurt by a discussion - it’s a good paper that does enough to survive an honest discussion of limitations.

I agree that this paper is theoretical enough to not need a discussion of societal impacts. Since the work “involves understanding deep neural networks”, I’m not sure that it “inherits the broader positive and negative societal impacts of deep learning”. It does very indirectly (which is what I think the authors are saying), but (also indirectly, but less so) it inherits more of the impacts of neural network interpretability than neural network application. One thing that isn’t touched is the societal impact of having good models of biological intelligence, which are huge and rarely discussed. I don’t think this warrants a flag for ethical review, since 1) there is so much ground between the state of the field and a societally impactful model and 2) I rarely see such discussion in comp neuro papers, but I do believe it warrants consideration if it is the ultimate goal.


**Main Review:**

Great paper! Impressively clear, well-scoped, doesn’t overpromise, delivers something interesting.

*Originality:* The work seems sufficiently original. It is a contribution to the community and is very clear and honest in comparing itself even to the most similar work, distinguishing itself specifically by focusing on learning rules. It includes a good range of related work and positions itself as a helpful addition.

*Quality:* This work seems technically sound. It scopes itself well and lists out certain claims, then goes on to provide experimental backing for each claim. The experiments are well-designed and clear, and answer the questions asked. Particular compliments to Fig 1, which illustrates the primary claim of the paper almost independently. Though some of this and other figures ask for visual inspection, they are backed up by descriptive statistics in the text. I have no qualms about experimentation quality. My only concern would be that I would have liked to see more tasks even of these complexities, so that there were more examples for various experiments (especially the ones using only N-CDI). Without these, the paper does seem to be putting all its eggs in one basket for certain claims, but the thoroughness of the experiments that are present mitigates this a lot.

The methods are intuitive and thorough. One concern is that some of the selected learning rules become incompatible with certain experiments, so it could be that a larger set of learning rules would have been valuable so that certain experiments weren’t limited to one or two rules, but I could see this being left for future work (and I realize that options are limited).

I don’t know that the paper discusses limitations sufficiently to satisfy the current NeurIPS guidelines (see below), but the paper also doesn’t overstate its contributions. I think it’s relatively intellectually honest. I do see it as a work in progress in that it leaves a lot of space for expansion, but as a paper in computational neuroscience I think it’s sufficient.

*Clarity:* This paper is really well-written. Many papers start off strong and proceed to fall apart writing-wise in the results, but this one has great organization, signposting, and writing throughout. It also follows the rule of “say what you’re going to say, say it, say what you said”, which was very useful for my understanding. My only concern is that I interpret the findings to be the following in order: “dynamics do not vary with learning rule (implicitly, for more complex tasks), representations vary with task complexity, dynamics do not vary with task complexity, dynamics do vary with learning rule under low input noise with simpler tasks, dynamics are confirmed to not vary with learning rule under low input noise with more complex tasks (suggested explanation being that more complex tasks have a more constrained solution space)” (quote from my summary). It’s a little confusing to understand how the beginning and end of this flow fit together. These aren’t exactly organized by either design choice (LR, input noise, etc.) or output (dynamics, representations) and might benefit from such an organization. I may also not fully understand the flow, so I am open to the existing ordering.

Minor: the paper starts off referring to task complexity as “task hyperparameters” exclusively; use of “task complexity” starts around halfway through the paper. Terms should be consistent regardless, but this is especially confusing because the paper is about learning rules. So, I recommend only using “task complexity” so no one has the opportunity to get confused even if they are skimming, even if the N in N-CDI could be considered a hyperparameter.

*Significance:* I think this paper is a useful contribution to understanding of how RNNs should be used when modeling biological intelligence. To my knowledge, the results are unique and set up both lots of ground for expansion and framework for experimentation meant to model the brain. To me, this seems like a worthwhile addition to the field.


**Time Spent Reviewing:**

6

---

> ### Author Response · Authors · 2021-08-10
> **Response**
>
> Thank you for your comments and suggestions on this paper.
>
> To your point on additional tasks:
> >We trained RNNs to perform the delayed non-match to sample (DNMS) task as described in [1]. We chose this task as it is a very commonly performed working memory task in animal experiments. We are still training more networks (particularly BPTT and FF) as of the posting of this response (10 Aug), but the current data paint a clear picture of conclusions consistent with the rest of the study. With 20 H and 20 GA networks, we observe in MDS plots that the representations strongly cluster, but the attractor structure MDS is overlapping. These results suggest individuality in representations and universality in attractor topology with different learning rules, consistent with our other tasks. We will add the completed figure as anew Appendix figure.
>
> To your point about increasing clarity:
> >Thank you for your comment that the paper is "really well-written" and for raising a point about the flow, which we believe you understand. To us, the biggest jump is in the transition from task complexity to looking at mechanisms under different input noises. We have amply considered how to organize these sections, but because different input noises and different task complexities have to be introduced before the final section (which combines both task input noise and complexity) we felt that this necessitated the ordering we used. After consideration about how we could make this more clear, we decided to incorporate a clearer roadmap at the beginning of the Results, with more connections about how these areas relate, as well as adjustments to the transitions between sections. We thank you for this feedback and hope this helps to increase clarity. Thank you for raising the inconsistent use of task hyperparameters and task complexity to our attention. We will change this language in the manuscript.
>
> To your point on societal impact:
> >we will remove the referenced limitation and incorporate additional text on the limitations you pointed out. We agree that, on an absolute scale, there is much ground to cover towards models of biological intelligence, and indeed we are focused on the more controlled neurophysiological settings where an RNN is used to model a single and simple task, and propose new hypotheses for the computation. In this respect, our goal is insight into neural computation, not making a model of biological intelligence.
>
> 1. Miconi T. Biologically plausible learning in recurrent neural networksreproduces neural dynamics observed during cognitive tasks.  Elife.  2017;6.doi:10.7554/eLife.20899

---

> > ### Comment · Reviewer_T4eL · 2021-08-30
> > **Thanks for the response**
> >
> > These additions sound good - for the additional tasks I think I'd need to see more to be able to judge accurately but that's fine since I think the paper is sufficient as it stands.
> >
> > Your note about the flow makes sense, I think that a clearer roadmap would be helpful to prime the reader for what they need to understand.
> >
> > Overall, I think the changes you've proposed make sense and don't significantly change my view of the paper (aside from the fact that I expect the writing to improve). Thanks for your effort!

---

### Official Review · Reviewer_obz4 · 2021-07-16

**Rating:** 6
**Confidence:** 3

**Summary:**

A popular paradigm for modeling the brain using task-optimized deep learning models identifies three key features governing the solutions learned by neural networks: architecture, task, and learning rule. Building on work by Maheswaranathan et al. [1], which showed that architecture influences the representations learned by trained recurrent neural networks (RNNs), but not their attractor structure, the authors of this work extend the analysis to study the effect of learning rule and task on the representations and attractor structure of trained RNNs — with broadly consistent results. The authors show that the attractor structure is largely invariant to the choice of learning rule (at least on two simple tasks: random dot motion and context-dependent integration, with large input noise), but the learned representations are not invariant, especially as the complexity of the task (the number of individual contexts to integrate) increases. However, the authors also exhibit a setting where the attractor structure is *not* invariant to the choice of learning rule: when the input noise is low in either task. They further show that the learned attractor structure depends sensitively on the amount of noise injected into the network, with a transition between one solution in the low-noise regime and another in the high-noise regime. While these results may seem contradictory, they may be consistent under the hypothesis that the attractor structure of trained RNNs exhibits universality across architectures and learning rules *when the task is sufficiently difficult*.

**Limitations And Societal Impact:**

The results are fairly presented and the authors are clear throughout the paper about the limitations of their work as well as potential future directions and impact on neuroscience.

**Main Review:**

If this is indeed a point the authors wish to make, it would have been nice to see the models trained on a greater diversity of tasks, with more interesting attractor topologies. [1] studied three tasks with fairly unique attractor topologies, which were further investigated and linearized. Of these the present work studies only the context dependent integration task, whose attractor topology has been studied exhaustively in previous works ([2], although the authors here identify a new solution as a function of input noise), and additionally studies a random dot motion task. But the only analysis of attractor structure reveals either one or two fixed points (for zero input) in each case. It is perhaps not so surprising that different learning rules converge on these same simple solutions (although it would be interesting to investigate how the attractor topologies differ in the case of low input noise! The authors could consider including this analysis). When the task is made more difficult by including N contexts, only two learning rules lead to successful training (backprop and a genetic algorithm). The two learning rules lead to consistent attractor structure as evaluated by the MDS analysis, but the attractor structure itself is not investigated.

This work includes several interesting conceptual insights beyond previous works, including the sensitive dependence of learned solutions on input noise. And the demonstration of settings where trained networks can exhibit universality to learning rules is an important step towards better understanding and interpreting neural networks, and using them as models of the brain. Overall the analyses are well performed and the presentation is clear. That said, the extensions this work presents beyond [1] are relatively modest, and no new methods or analysis techniques are introduced which could be of independent interest (aside from the silhouette score, the methods in the present work appear to be identical to those introduced in [1]).

Finally, a “cautionary tale” discussed in [1] is that CCA may actually be a poor metric of representational similarity, given that in some settings the canonical correlation between trained and untrained RNNs can be greater than the canonical correlation between two trained RNNs with different design choices (e.g. nonlinearities). It would be interesting for the authors here to discuss this concern, and/or to try out a different metric for representational similarity.



[1] N. Maheswaranathan, A. H. Williams, M. D. Golub, S. Ganguli, and D. Sussillo, “Universality and individuality in neural dynamics across large populations of recurrent networks,” 2019.

[2] H. F. Song, G. R. Yang, and X. J. Wang, “Training excitatory-inhibitory recurrent neural networks for cognitive tasks: a simple and flexible framework,” PLoS Comput. Biol., vol. 12, no. 2, pp. 1–30, 2016.

**Time Spent Reviewing:**

7

---

> ### Author Response · Authors · 2021-08-10
> **Response**
>
> Thank you for your comments and suggestions on this paper.
>
> To your point on it being nice to add additional tasks:
> >Thanks for your comment regarding additional tasks.  We have taken steps to add an additional task comparison.  We wanted to highlight that our priority in task design was to (1) look at a very well-studied task in the neuroscience literature to assess how learning rule and task complexity affected mechanism, and (2) keep all other design choices aside from learning rule as similar as possible.  We therefore choose to investigate integration based tasks of similar structure since these tasks have been well studied and enable us to maintain similar task designs in a controlled fashion, while easily varying complexity and noise. Although we believe the current family of tasks and analyses support our conclusions, in response to this point, we decided to incorporate a completely different neurophysiological task as an Appendix Figure, to show similar trends extend to a different type of task (more individuality in representations, but universality in attractor topology).
>
> >We trained RNNs to perform the delayed non-match to sample (DNMS) task as described in [2]. We chose this task as it is a very commonly performed working memory task in animal experiments. We are still training more networks (particularly BPTT and FF) as of the posting of this response (10 Aug), but the current data paint a clear picture of conclusions consistent with the rest of the study. With 20 H and 20 GA networks, we observe in MDS plots that the representations strongly cluster, but the attractor structure MDS is overlapping. These results suggest individuality in representations and universality in attractor topology with different learning rules, consistent with our other tasks. We will add the completed figure as a new Appendix figure.
>
> To your point about exploring the attractor structure for low noise tasks:
> >Thank you for raising these points. We think this is a great point and we have since added additional panels exploring the variations in attractor topology between different learning rules in the case of low input noise. While this topic is touched on in figure 3, where the low noise attractor structure is shown for example BPTT networks, we will include additional plots showing the attractor structure in the case of low noise for the GA, Hebbian, and FF trained networks. In short, these plots show a single line attractor without multiple attractors for the same input, reflecting the transient integration mechanism we described in Figure 4. We note some subtle differences in the curvature of this line attractor between different learning rules, which likely leads to the high silhouette score of 0.45 seen in Figure 3(b).
>
> To your point about the attractor structure itself not being investigated for the 3+ CDI tasks:
> >This will be clarified in the text and an additional figure will be added to the appendix depicting example attractor structures between BPTT and GA networks for the more complex tasks. Ultimately, we find the attractor structure is similar to that of the CDI task just with additional contexts.  As additional dimensions are added to the task, the network instantiates additional line attractors for each context.
>
> To your point about the extensions beyond Maheswaranathan et al. [3] being relatively modest:
> >We purposefully chose to use the same metrics as in [3] (to facilitate cleaner and direct comparisons to prior work)but we applied them to very conceptually different questions. We emphasize that the hypotheses we addressed are absolutely distinct from Maheswaranathan and colleagues [3], and we believe the conclusions of our study will be of general interest to computational neuroscientists. Specifically, we looked at key factors of variation in neuroscience studies not addressed by [3], and also showed how they could result in different conclusions in published work. We believe the reviewer understands this (as you highlighted our conceptual insights), but to be absolutely clear, we wish to emphasize that:
> * We looked at how the learning algorithm affects the mechanism, something not explored by [3], which used BPTT for all architectures.
> * We used the body of N-CDI tasks to answer questions about how task complexity affects representation and attractor similarity, something not explored by [3].
> * We found a design variable, input noise, that strongly affects attractor structure, something not explored by [3]. Further, we showed how this could result in different integration mechanisms (Figure 4), which has significance for neurophysiological experiments and hypotheses (e.g., continuous vs discrete integration).This shows that convergence in attractor topology is not due to a trivial lack of alternative solutions.
> * We found that task noise is less of a concern when the task is sufficiently complex.
> Thus,  while the analyses are purposefully chosen to be similar,  the primary extension of our study are the conceptual questions we tackled with respect to learning rule, task input noise, and task complexity.  These questions are of direct interest to the neuroscience community, and especially to modeling approaches that use RNN to model neurophysiological data. We have updated the Discussion to make this more clear.
>
> To your points about being cautious about the use of singular vector canonical correlation analysis:
> >Thank you for raising this point. We have added a brief discussion of this concern to the manuscript. In response to this concern, we performed additional analysis comparing the representational geometries before and after training RNNs. We find that for the RDM task, trained and untrained networks are less similar than networks trained with different learning rules. Specifically, we observe networks performed to train the RDM task exhibit different representations before and after training (BPTT silhouette: 0.57, GA: 0.60, H: 0.18, FF: 0.48). This is much higher than after training (silhouette 0.05 across all rules, Figure 1 in the text).
>
> 1. Sussillo D, Barak O. Opening the black box: low-dimensional dynamics in high-dimensional recurrent neuralnetworks. Neural Comput. 2013;25: 626–649.
> 2. Miconi T. Biologically plausible learning in recurrent neural networks reproduces neural dynamics observedduring cognitive tasks. Elife. 2017;6. doi:10.7554/eLife.20899
> 3. Maheswaranathan N, Williams A, Golub M, Ganguli S, Sussillo D. Universality and individuality in neuraldynamics across large populations of recurrent networks. In: Wallach H, Larochelle H, Beygelzimer A, d’Alché-BucF, Fox E, Garnett R, editors. Advances in Neural Information Processing Systems 32. Curran Associates, Inc.;2019. pp. 15603–15615.

---

> > ### Comment · Reviewer_obz4 · 2021-09-10
> > **Thanks for your response**
> >
> > Thanks for your response, and for the clarifications and additional analyses -- these have addressed a number of my concerns, and I have raised my score to marginally above the acceptance threshold. Are there any updates on the experiments with new tasks?

---

> > > ### Author Response · Authors · 2021-09-16
> > > **New task updates**
> > >
> > > Thank you again for your feedback.
> > >
> > > Yes, we have more results in progress using the delayed non-match to sample (DNMS) task as described in [1]. We are now able to successfully train RNNs with all four learning rules (BPTT, GA, H, and FF) on the DNMS task.  We are currently training more and plan to have at least 20 trained with each learning rule in the final analysis.
> > >
> > > In results with 10 networks for each learning rule, RNNs instantiate a similar attractor structure to solve the task (attractor MDS silhouette score: 0.09, figure link: https://imgur.com/a/SsNgTVr). This is consistent with our prior results showing similar attractor structures across learning rules on the RDM and CDI based tasks.
> > >
> > > 1. Miconi T. Biologically plausible learning in recurrent neural networks reproduces neural dynamics observed during cognitive tasks. Elife. 2017;6. doi:10.7554/eLife.20899

---

### Official Review · Reviewer_MbCb · 2021-07-17

**Rating:** 7
**Confidence:** 5

**Summary:**

This paper studies the effect of learning rules on the algorithms learned by recurrent networks on context-dependent integration tasks. A number of continuous time vanilla RNNs are trained on integration tasks with different learning rules: backprop, genetic algorithms, hebbian learning, and full-force. The paper finds that across these learning rules, all networks learn consistent attractor topologies in order to solve the task. As the complexity of the task is increased (increasing number of contexts), the representations start to differ, but the attractor structure is preserved. The paper finds that using line attractors to solve integration seems to be a universal property of these networks, independent of learning rule. Finally, the paper shows how properties of the learned attractors (both the topology and whether the attractors are continuous manifolds or discrete points) vary with task input noise.

**Main Review:**

Overall, I enjoyed reading this paper. It is well written and addresses an important problem: what aspects of the algorithms learned by networks trained to solve particular tasks are sensitive to the choice of learning rule used to train the network?

### Questions / concerns
- The tasks are presented as being a part of three separate types (RDM, CDI, and N-CDI), but I feel like it makes more sense to present the tasks as all being part of the same family of tasks. They are all context-dependent integration tasks, but what varies is the number of contexts: ranging from one (RDM) to two (CDI) to multiple (N-CDI). Is my understanding correct?
- L184/185: "This is possibly due to the fact that Hebbian RNNs do not compute a linear readout ... but instead use an arbitrary recurrent neuron as the output". This seems like a significant difference, and should be controlled for. Why not train the other networks (BPTT / GA) in this fashion, by picking an arbitrary neuron to be the readout (as opposed to having a linear readout). That would help control for this difference.
- It would be nice to see results for other tasks (besides integration). Are there other simple tasks that might be good candidates?
- I thought the results showing a transition from transient to stable integration with increasing task noise particularly interesting. Are there other aspects of the task that might induce similar transitions? This point is particularly relevant for neuroscientists that might observe continuous or discrete integration in the brain.
- In Figure 2a, it looks like the BPTT representations are all similar, all the GA networks are different (not only from BPTT, but different amongst themselves as well). Any intuition for this? It looks like some GA networks are similar to the BPTT networks, especially for smaller numbers of contexts. Is that variability due just to the random seed? Can you identify what it is about certain random seeds that makes some GA networks look like BPTT networks?

**Time Spent Reviewing:**

3

---

> ### Author Response · Authors · 2021-08-10
> **Response**
>
> Thank you for your comments and suggestions on this paper.
>
>
> To your question on if your understanding of whether the tasks are in the same family:
> >We agree with your understanding that these tasks comprise a family. We view these tasks as distributed along a spectrum of task complexity, with increasing inputs as there are more contexts. We do view that from RDM to CDI, there is an important qualitative change in that the network receives a static context input that is not present in the RDM task(as opposed to a “1-CDI” task, where the network would receive a fixed context input). The addition of the context signal from RDM to CDI means the network can’t directly use a simple integration strategy, but must selectively choose an input to attend to, as documented in [1]. We will update the manuscript to more clearly describe this.
>
>
> To your point on controlling for the Hebbian output:
> >Thank you for raising this point.  We were initially motivated by studying learning rules in the literature and implementing them as described (for Hebbian, see[2]), but we agree with your point on controlling for the difference in output configuration. We therefore retrained additional BPTT and GA networks to perform the RDM and CDI tasks using a single recurrent unit as output to control for this difference, as suggested. We find that attractor topologies remain highly overlapping as originally reported in the paper (RDM silhouette score: -0.03, CDI silhouette score: 0.1). Additionally, when BPTT and GA trained networks use a recurrent unit as output, the Silhouette scores of the representations across BPTT, GA, and Hebbian trained networks remains low (RDM silhouette score: -0.051, CDI silhouette score: 0.057). This control is therefore consistent with the conclusions of the paper. We will include the results of this analysis as an Appendix figure.
>
> To your point on other simple candidate tasks:
> >We initially designed the tasks to be distributed along a spectrum of complexity such that we could examine the effect of task complexity on dynamical mechanism and representation with minimal other confounds. We therefore aimed to keep all other design choices aside from learning rule as similar as possible, focusing on integration mechanisms. This enabled us to maintain similar task designs in a controlled fashion, while varying complexity and noise. Although we believe the current family of tasks and analyses support our conclusions, in response to this point, we decided to incorporate a completely different neurophysiological task as an Appendix Figure, to show similar trends extend to a different type of task (more individuality in representations, but universality in attractor topology).
>
> >We trained RNNs to perform the delayed non-match to sample (DNMS) task as described in [2]. We chose this task as it is a very commonly performed working memory task in animal experiments. We are still training more networks (particularly BPTT and FF) as of the posting of this response (10 Aug), but the current data paint a clear picture of conclusions consistent with the rest of the study. With 20 H and 20 GA networks, we observe in MDS plots that the representations strongly cluster, but the attractor structure MDS is overlapping. These results suggest individuality in representations and universality in attractor topology with different learning rules, consistent with our other tasks. We will add the completed figure as a new Appendix figure.
>
> To your point on other aspects that might induce transitions:
> >Based on Appendix Figure 7, we also observed that different activation functions (tanh or relu) could result in different integration strategies for the RDM but not the CDI task. In the case of the CDI task, tanh and relu activations both resulted in the stable integration strategy. Together, this leads us to speculate that increasing task difficulty, whether through task noise (Figure 3),or potentially task complexity (Appendix Figure 7, and observation that CDI uses stable integration irrespective of whether the activation is tanh or relu), may lead to stable integration. We believe this is an interesting point to highlight, and will add this observation and speculation to the discussion. We agree this would be particularly relevant for neuroscientists.
>
> To your point on variability in GA due to random seed:
> >We agree with your observation, and that there is greater variance in general for the GA networks in representation (though not attractor structure).  The only difference in these GA networks is the random seed, as the GA algorithm to train all these networks was identical. At the moment, we have not identified any structure in random seeds that causes some GA networks to look like BPTT networks, but agree this is an interesting question. We believe this is more fundamentally related to the GA algorithm research, and would be an interesting avenue for future work in general machine learning (that is, if it’s possible to select certain seeds such that the GA learning trajectory converges similarly to BPTT).
>
> 1. Mante V, Sussillo D, Shenoy KV, Newsome WT. Context-dependent computation by recurrent dynamics inprefrontal cortex. Nature. 2013;503: 78–84.
> 2. Miconi T. Biologically plausible learning in recurrent neural networks reproduces neural dynamics observedduring cognitive tasks. Elife. 2017;6. doi:10.7554/eLife.20899

---

> > ### Comment · Reviewer_MbCb · 2021-08-30
> > **Thank you for your response**
> >
> > Thank you for your response. I am curious what results you will see on additional tasks. Varying the number of contexts within CDI does not feel like sufficiently diverse enough variation to give me confidence that you will see the same general pattern of results on totally different types of tasks.

---

### Decision · Program_Chairs · 2021-09-27

**Decision:**

Accept (Poster)

**Comment:**

This paper is an empirical study of the impact of learning rule and input noise on the RNN solutions in a small family of tasks. The reviewers agree that writing is clear, and showed excitement in seeing the results, but concerned with the narrow range of tasks. Reviewers made excellent constructive suggestions  that I strongly encourage the authors to incorporate in the final, stronger manuscript.